

# Identification of 10 Hub genes related to the progression of colorectal cancer by co-expression analysis

Jie Meng[1,2], Rui Su[2], Yun Liao[2], Yanyan Li[2] and Ling Li[2]

[1] Department of Pharmacy, Anhui University of Chinese Medicine, Hefei, China
[2] Department of Pharmacy, Tongren Hospital, Shanghai Jiaotong University School of Medicine, Shanghai, China

## ABSTRACT

**Background:** Colorectal cancer (CRC) is the third most common cancer in the world. The present study is aimed at identifying hub genes associated with the progression of CRC.

**Method:** The data of the patients with CRC were obtained from the Gene Expression Omnibus (GEO) database and assessed by weighted gene co-expression network analysis (WGCNA), Gene Ontology (GO) and Kyoto Encyclopedia of Genes and Genomes (KEGG) enrichment analyses performed in R by WGCNA, several hub genes that regulate the mechanism of tumorigenesis in CRC were identified. Differentially expressed genes in the data sets GSE28000 and GSE42284 were used to construct a co-expression network for WGCNA. The yellow, black and blue modules associated with CRC level were filtered. Combining the co-expression network and the PPI network, 15 candidate hub genes were screened.

**Results:** After validation using the TCGA-COAD dataset, a total of 10 hub genes (MT1X, MT1G, MT2A, CXCL8, IL1B, CXCL5, CXCL11, IL10RA, GZMB, KIT) closely related to the progression of CRC were identified. The expressions of MT1G, CXCL8, IL1B, CXCL5, CXCL11 and GZMB in CRC tissues were higher than normal tissues ($p$-value < 0.05). The expressions of MT1X, MT2A, IL10RA and KIT in CRC tissues were lower than normal tissues ($p$-value < 0.05).

**Conclusions:** By combining with a series of methods including GO enrichment analysis, KEGG pathway analysis, PPI network analysis and gene co-expression network analysis, we identified 10 hub genes that were associated with the progression of CRC.

Corresponding author
Ling Li, LL2699@shtrhospital.com

## INTRODUCTION

Colorectal cancer (CRC) is the third most common cancer in the world and considered as the second leading cause of cancer-associated deaths (*Brody, 2015*). Due to lack of early specific disease symptoms, it is often identified in advanced stages leading to poor prognosis (*Simon, 2016*). Multiple biomarkers that can help in improving the diagnosis and treatment monitoring have been identified for CRC (*Lech et al., 2016*;

*Okugawa, Grady & Goel, 2015*). It has been reported that *THBS2* can serve as a prognostic biomarker and also the expression of *THBS2* is significantly associated with lymphatic invasion and TNM staging of CRC patients (*Wang et al., 2016*). The overexpression of *CHD4* induced microsatellite instability-high (MSI-H) colorectal cell (CRC) radio-resistance is also reported. The knockdowns of CHD4 enhances radio-sensitivity in microsatellite stabilization especially in CRC (*Wang et al., 2019*). Few studies have shown that the immune system also plays an important role in the development of CRC (*Becht et al., 2016*; *Yin et al., 2017*). As can be seen that the mechanism of CRC is complicated and multifaceted in nature, it requires further exploration of mechanisms for the occurrence and development of CRC.

Advances in the sequencing technologies have provided excellent tools and platforms for cancer research including CRC (*Srivastava, Mangal & Agarwal, 2014*). By correlating the clinical data with molecular mechanisms, new biomarkers for diagnosis, treatment, and prognosis can be restored. Microarray could be used to probe the key biomarkers and provide a better understanding of the molecular mechanisms involved in CRC. Until now, clinically applicable biomarkers are still lacking. Therefore, exploring novel and effective molecular biomarkers to elucidate effective therapeutic targets for CRC patients is still imperative. In this study, we focused on the different expression pattern between the CRC tumor tissues and matched normal tissues. To discover the hub genes and key pathways associated with the initiation and progression of CRC, we applied differential gene expression analysis and functional enrichment analysis. Recently it was demonstrated that the genetic characteristics of the ER and PR pathways can serve as a new marker for CRC prognosis and management (*Liu, 2016*). Weighted gene co-expression network analysis (WGCNA) is an R-package for weighted correlation network analysis and can be used as a data exploration tool for genetic screening (sorting) to find clusters (modules) of highly related genes. It has been widely used to find hub genes in various cancers. For example, studies using WGCNA and UBE2S to identify 11 gene co-expression clusters from large-scale breast cancer data suggest poor prognosis in breast cancer (*Clarke et al., 2013*). In order to explore the progression of CRC, we have used this algorithm to identify hub genes associated with clinical features.

In the current study, by combining GO enrichment analysis, KEGG pathway analysis, PPI network analysis, gene co-expression network analysis and other bioinformatic methods, we aim to identify the potential key hub genes and modules involved in CRC initiation and progression. In conclusion, we identified 10 hub genes that participated in several cancer-relevant pathways and their abnormal expression are correlated with the clinical progression and prognosis of CRC people by overall survival analysis.

## MATERIALS AND METHOD

### Data download and processing

GSE28000 and GSE42284 were obtained from NCBI Gene Expression Omnibus (GEO). The GSE28000 has a total of 115 samples, the platform was Agilent-014850 Whole Human Genome Microarray 4 × 44 K G4112F. GSE42284 was consist of 188 diseased samples,

the platform was Agilent *Homo sapiens* 37 K DiscoverPrint_19742. We used batch normalization to correct the two-chip data and all data are normalized.

### Screening of differentially expressed genes

Difference analysis of 34 normal and 269 CRC samples was performed using the limma package, $|logFC| > 1$, $p$-value $< 0.05$ was defined as differentially expressed genes (DEGs).

### GO enrichment analysis and KEGG pathway analysis

A total of 694 upregulated and 1,271 downregulated genes were analyzed by Gene Ontology (GO) enrichment and Kyoto Encyclopedia of Genes and Genomes (KEGG) pathway using the R package cluster profiler. $P$-value $< 0.05$ was defined as a meaningful enrichment analysis result. The KEGG pathway was analyzed by R package with a threshold $p$-value $< 0.05$. GO and KEGG pathway analysis was used to predict potential functions.

### Gene co-expression network analysis

The gene co-expression network was constructed using the WGCNA package. The top 25% of the genes from the variance plot were screened for constructing a weighted co-expression network. The network module was segmented using a dynamic cut tree algorithm. In order to test the stability of each identified module randomly divided training and test set were generated using the preservation function module stability in the WGCNA package. Correlation between modules and clinical features was assessed by Pearson correlation testing to search for key modules. Clinical information included gender, age, race, stage, location for WGCNA analysis.

### Hub gene screening

Genes in the overlapping region were selected after combining the DEGs and the module genes. These were obtained from the string database. The PPI network was constructed and a higher degree node in the network was selected as the hub genes.

### Data set verification

In order to verify if the gene expression significantly associated with CRC, we validated 15 candidate hub genes using TCGA-COAD data in GEPIA database.

### Hub gene survival analysis

Kaplan–Meier—Plotter database (http://kmplot.com/analysis/) in CRC data set were used for survival analysis.

## RESULTS

### Acquisition of microarray data

A total of 303 samples and 18,588 genes (Table S1) in GSE28000 and GSE42284 were corrected by batch normalization. File annotation information from Agilent-014850 Whole Human Genome Microarray 4 × 44 K G4112F (Feature Number version) and Agilent *Homo sapiens* 37 K DiscoverPrint_19742 platforms is shown in (Table S2).

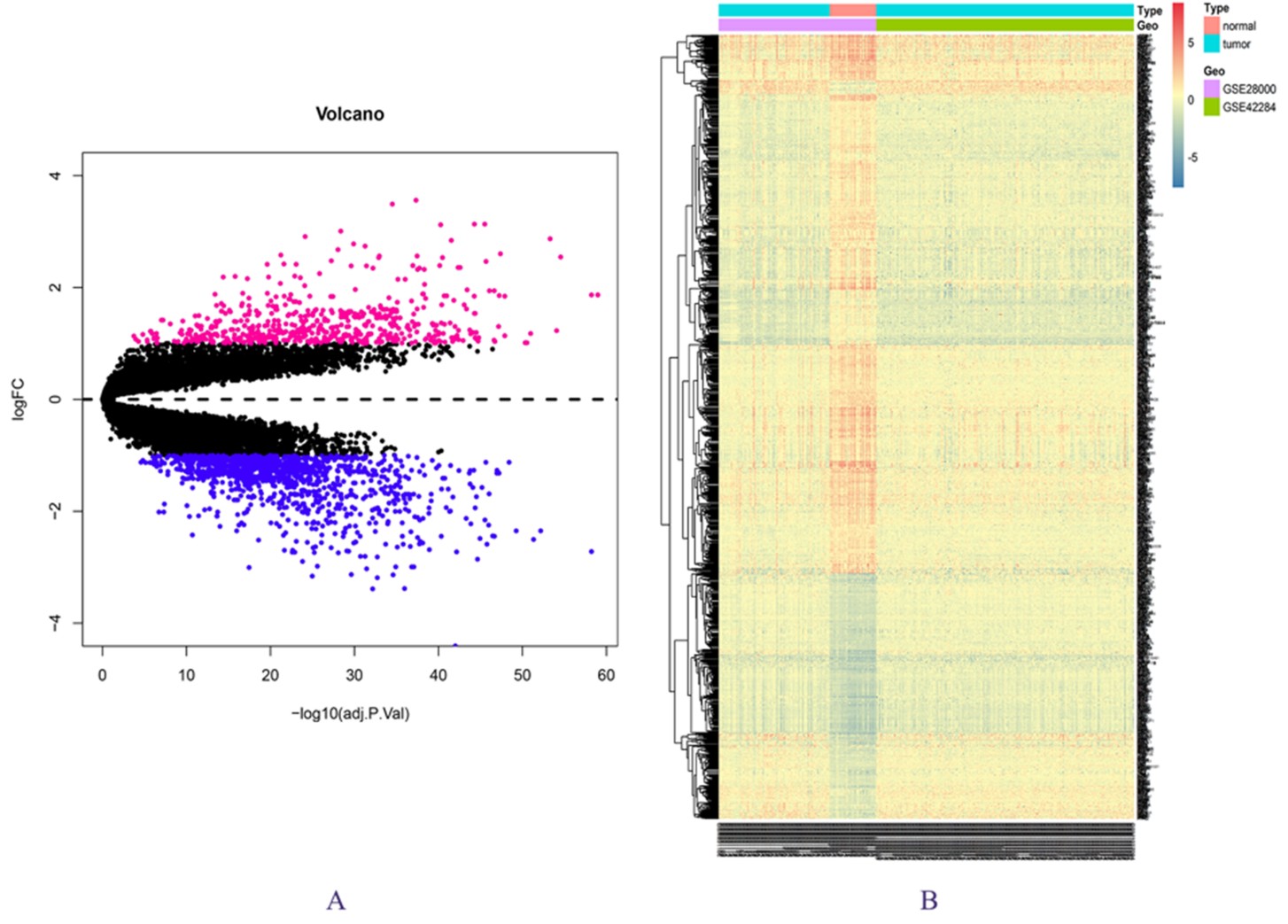

**Figure 1 DEGs in GSE28000 and GSE42284.** (A) Volcanic map of gene expression values between colorectal cancer tissues and normal tissues. Vermilion is the up-regulated gene and blue is the down-regulated gene. (B) Differential gene heat map.

## DEGs screening

The expression matrix data from 34 normal and 269 cancer samples were compared and analyzed using the limma package. By employing $p$-value < 0.05 and |logFC| ≥ 1 as critical criteria, a total of 1,041 DEGs were obtained (Fig. 1A; Table S3). DEGs are shown in the volcano and the heat map (Figs. 1A and 1B).

## GO enrichment analysis and KEGG pathway analysis results

A total of 694 up-regulated and 1,271 down-regulated genes were analyzed by GO enrichment using the R package cluster profiler. $P$-value < 0.05 was defined as a meaningful enrichment analysis result (Fig. 2A; Table S4). As shown in Fig. 2A, 1,965 genes were significantly enriched in receptor ligand activity, G protein-coupled receptor binding and cytokine activity. KEGG pathway enrichment analysis results were shown in

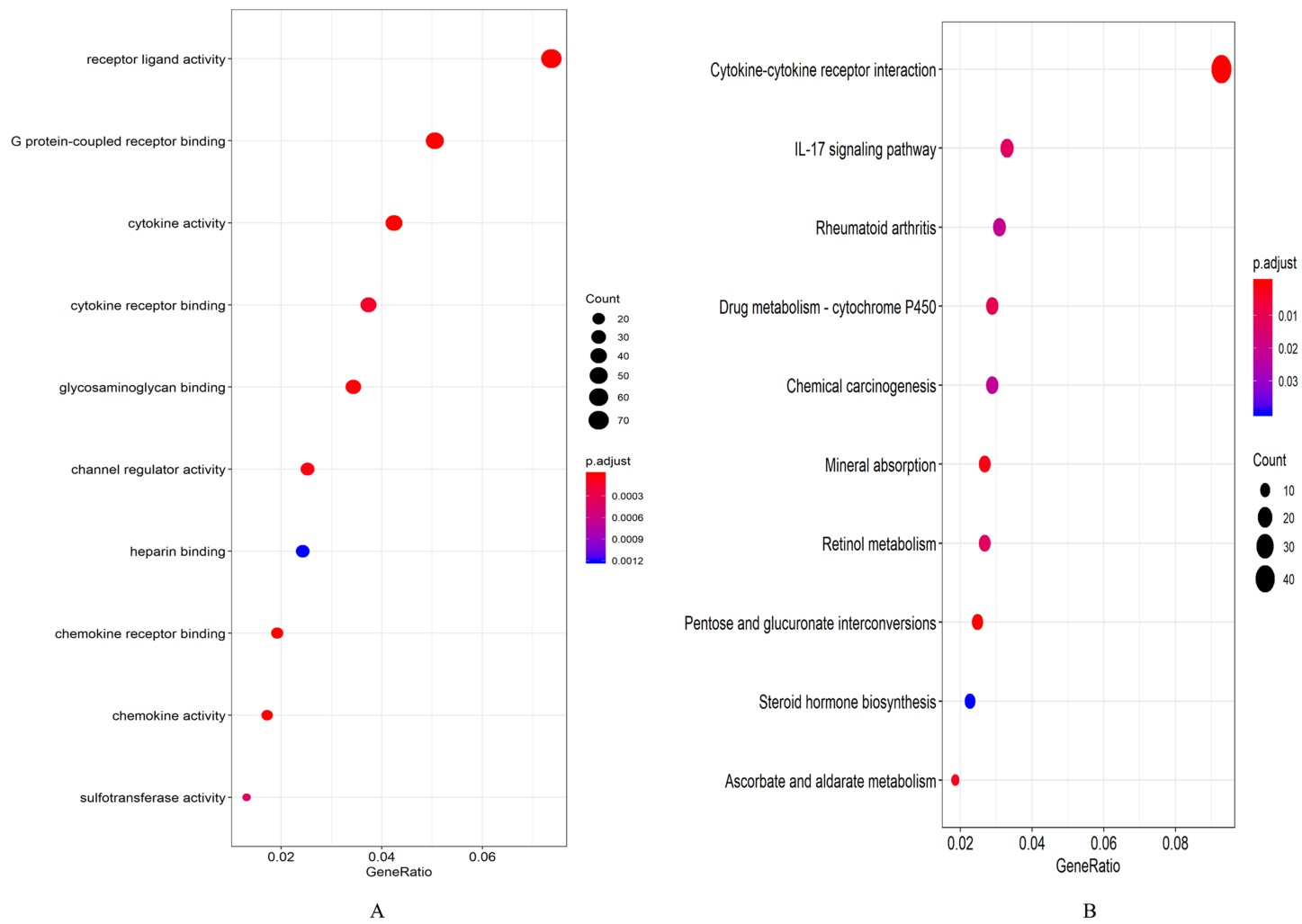

**Figure 2 Enrichment analysis results.** (A) GO enrichment analysis results of DEGs. (B) Results of the first 12 KEGG pathways in enrichment analysis of DEGs.

Fig. 2B and Table S5, which were significantly enriched in cytokine–cytokine receptor interaction.

## Gene co-expression network analysis results

GSE28000 and GSE42284 consisted of a total of 303 samples and 18,588 genes. The top 25% of the genes in the variance plot were screened to construct a weighted co-expression network. In order to ensure the reliability of the network structure 42 outliers were removed (Fig. 3A). A total of 11 modules (Figs. 3C and 3D) were obtained by selecting an appropriate soft threshold power = 9 (Fig. 3B) according to the scale-free network. Further, to evaluate the stability of each identified module, the training (Table S6) and test set (Table S7) were randomly divided and the module stability was calculated using the module preservation function (permutations = 200) in the WGCNA package. The clinical information related to GSE26712 is shown in Table S8. The yellow, black and blue modules

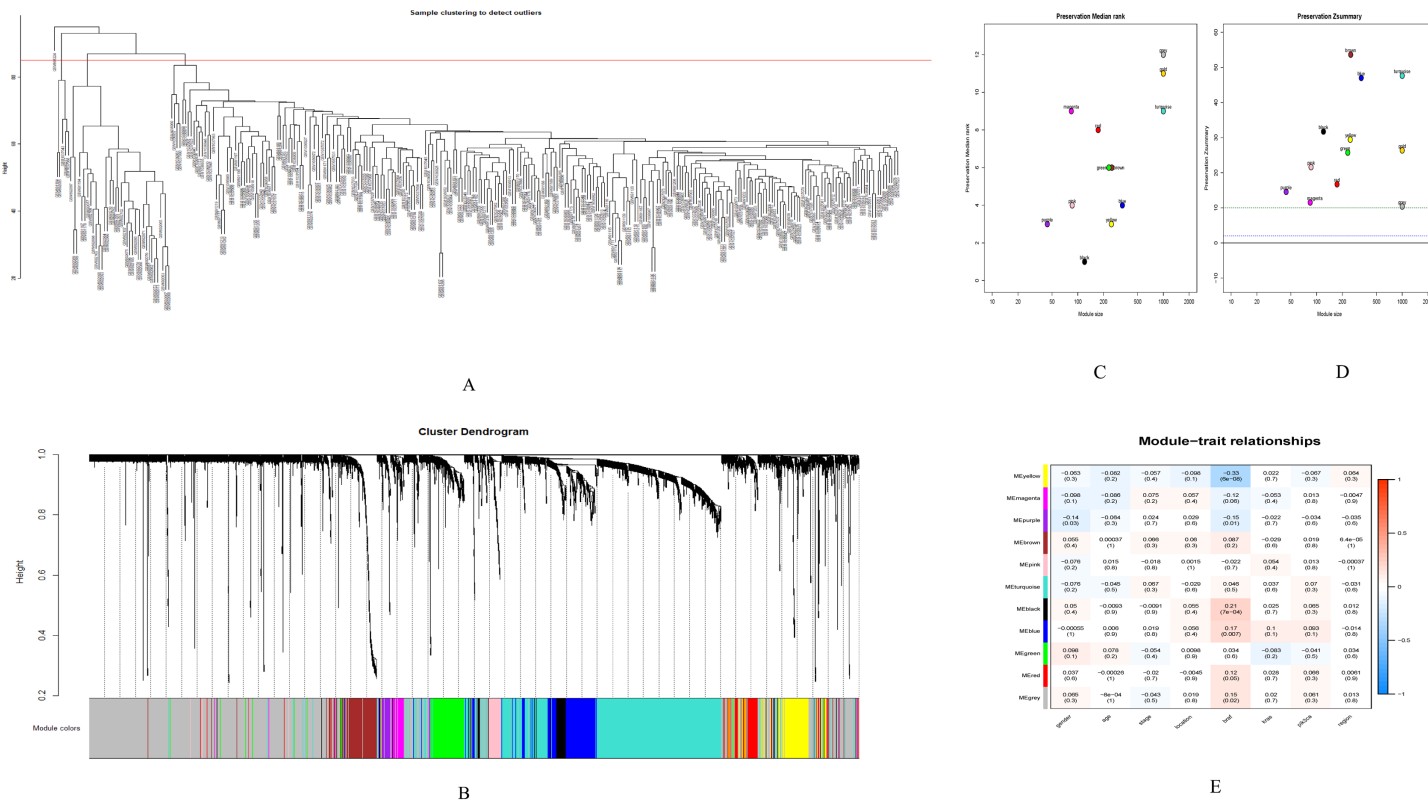

**Figure 3 WGCNA.** (A) Sample cluster analysis. (B) The top panel represents gene tree and the bottom a gene module with different colors. (C) Relationship between the two preservation statistics. (D) Module preservation as a function of module quality. (E) Correlation between modules and features. The upper number in each cell is the correlation coefficient between the clinical features and each module, and the lower number is the corresponding *p*-value. Module size and preservation scores are shown in the *x* and *y* axes, respectively. Module numbers are shown next to the circles. Modules with Z summary scores >10 (above the red dotted line) are considered highly preserved, Z summary scores between 2 and 10 (between the blue and red dotted lines) are weak to moderately preserved, and Z summary scores <2 (below the blue dotted line) are not preserved.

that are significantly associated with clinical features were selected as candidate key modules (Fig. 3E).

## Screening of hub genes

The yellow, black and blue modules were respectively intersected with the differential genes, and the genes in the intersection region are shown in Figs. 4A–4C.

The results of gene enrichment analysis of the intersection of DEGs and yellow, black and blue modules are shown in Figs. 5A–5F.

## Survival analysis of hub genes

PPI network analysis was performed on the genes from the intersection regions, and the interacting proteins with confidence >0.7 was selected. The yellow module had 27 nodes and 33 edges, the black module had 19 nodes and 73 edges, and the blue module had 18 nodes, 20 sides (Figs. 6A–6C).

The degree of each node in the network based on Cytoscpe was calculated. The top five nodes in the degree ranking in the module network were selected as candidate hub genes.

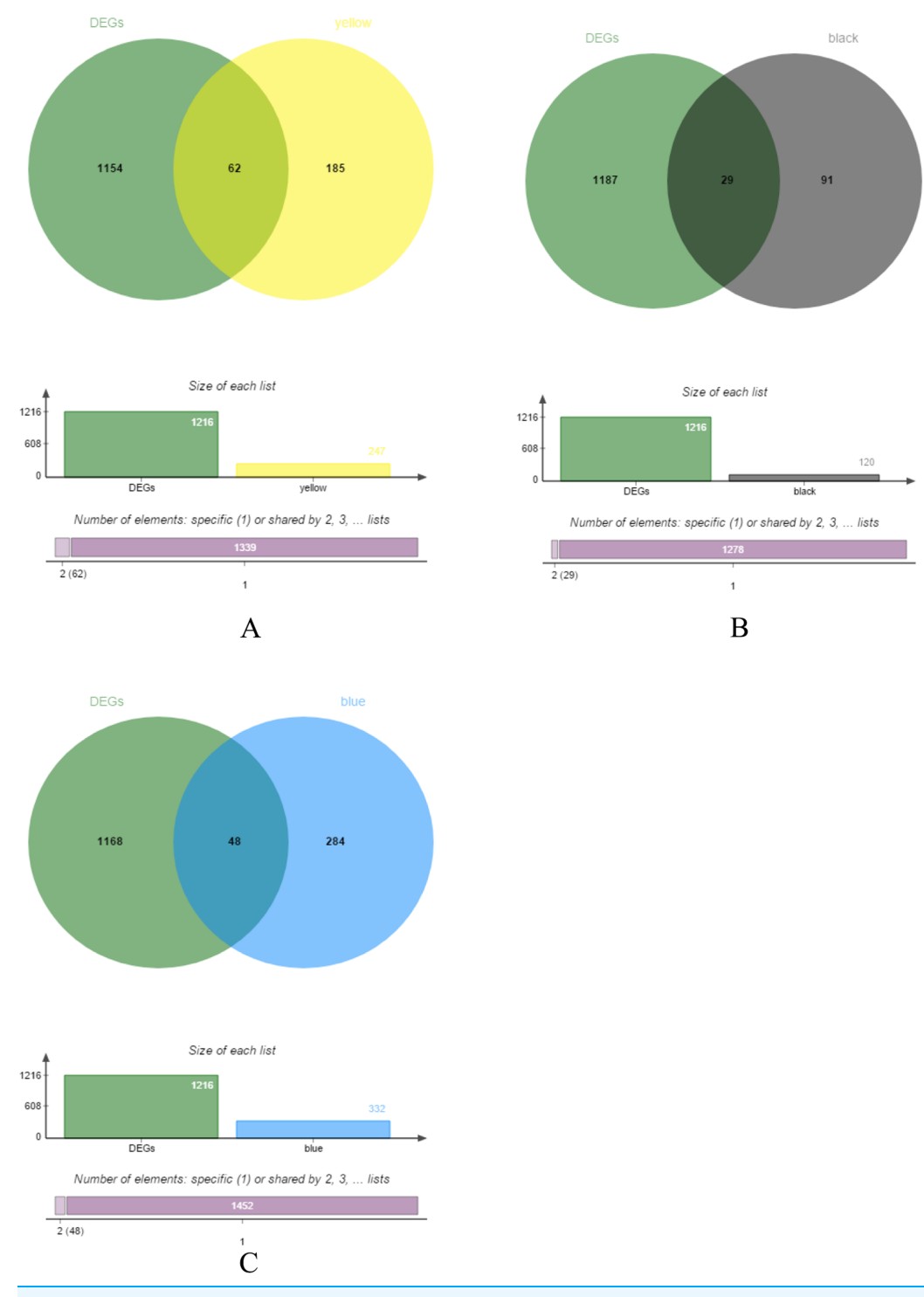

**Figure 4 DEGs and module gene Venn diagram.** (A) DEGs and yellow module gene Venn diagram. (B) DEGs and black module gene Venn diagram. (C) DEGs and blue module gene Venn diagram.

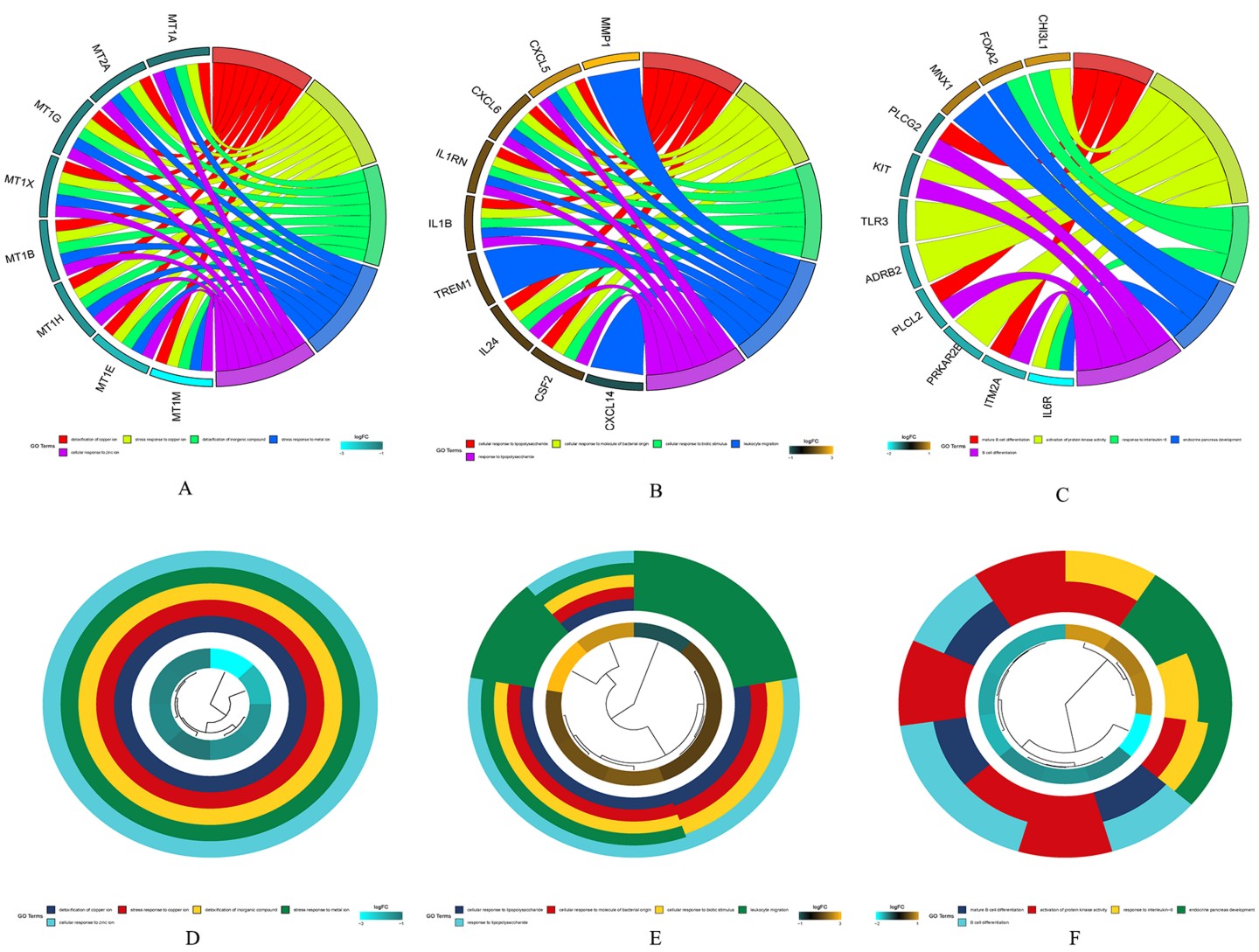

**Figure 5** **The results of gene enrichment analysis of the intersection of DEGs and yellow, black and blue modules.** (A) Gene enrichment analysis results of the intersection region of DEGs and yellow modules. (B) Gene enrichment analysis results of the intersection region of DEGs and black modules (C) Gene enrichment analysis results of the intersection region of DEGs and blue modules. (D) Yellow module GO enrichment analysis results. (E) Black module GO enrichment analysis results. (F) Blue Yellow module GO enrichment analysis results.

A total of 15 candidate hub genes were identified. The top five nodes in the yellow module were *MT1H, MT1X, MT1E, MT1G, MT2A*, and the top five nodes in the black module were *CXCL8, CSF2, IL1B, CXCL5, IL1A*, respectively. The top five nodes in the blue module were *TLR3, CXCL11, IL10RA, GZMB* and *KIT*.

## Data set verification

Similar results were obtained for the hub genes screened by the TCGA-COAD data validation in Figs. 7A–7J, suggesting reliable findings of the present study. A total of 15 candidate hub genes were verified, among these 10 genes: *MT1X, MT1G, MT2A, CXCL8,*

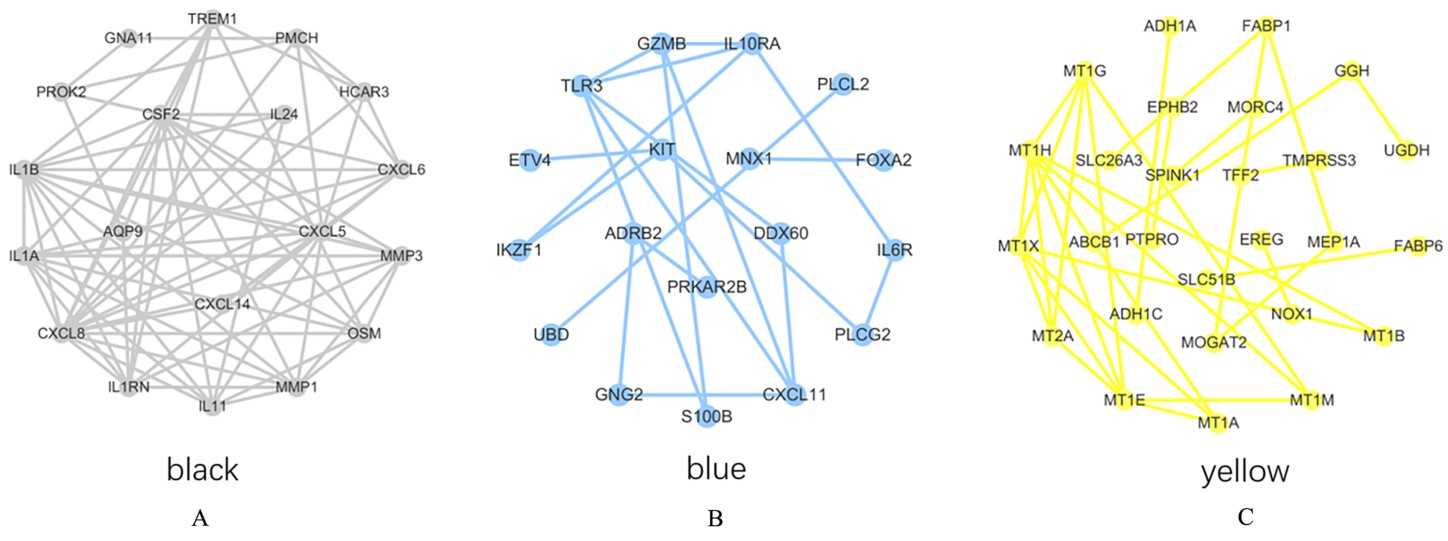

**Figure 6** **PPI network analysis results.** (A) Black module. (B) Blue module. (C) Yellow module. 

*IL1B, CXCL5, CXCL11, IL10RA, GZMB, KIT* were significantly different between the normal group and the cancer group.

## Survival analysis

Survival analysis of the hub genes selected above was performed in Figs. 8A and 8B. Findings suggest that two genes were significantly associated with the prognosis of patients (*p*-value < 0.05), which is *IL10RA* and *KIT*. With the increase of IL10RA and KIT expression, the total survival time was significantly prolonged.

## DISCUSSION

Over the past few decades, the mortality rates associated with CRC has increased due to the higher incidence in the young population (*Connell et al., 2017*; *The Lancet Oncology, 2017*). Long term research in CRC has led to many substantial advances in the diagnostic and therapeutic techniques. For example, effective prognostic biomarkers for CRC are CEA levels, circulating tumor DNA (ctDNA), MS instability and certain genetic characteristics (*Duffy, 2015*). Analysis can be used to diagnose, identify and track tumor-specific changes associated with disease progression and to guide treatment decisions (*Osumi et al., 2019*). miRNAs can also be used as diagnostic and prognostic biomarkers for assessing tumor development, progression, invasion, metastasis and reaction to chemotherapeutic drugs (*Shirafkan et al., 2018*). It can also help in the early identification of differences between responder and non-responder individuals (*Ballester, Rashtak & Boardman, 2016*). Advances in bioinformatics and genetics have led to the development of biomarkers and genetic models that can help to select responders and to assess prognosis, and thereby rationalizing, individualizing and improving the prognosis.

In this study, we identified hub genes associated with CRCs and established a gene co-expression network analysis. Initially, most of the response genes demonstrated a

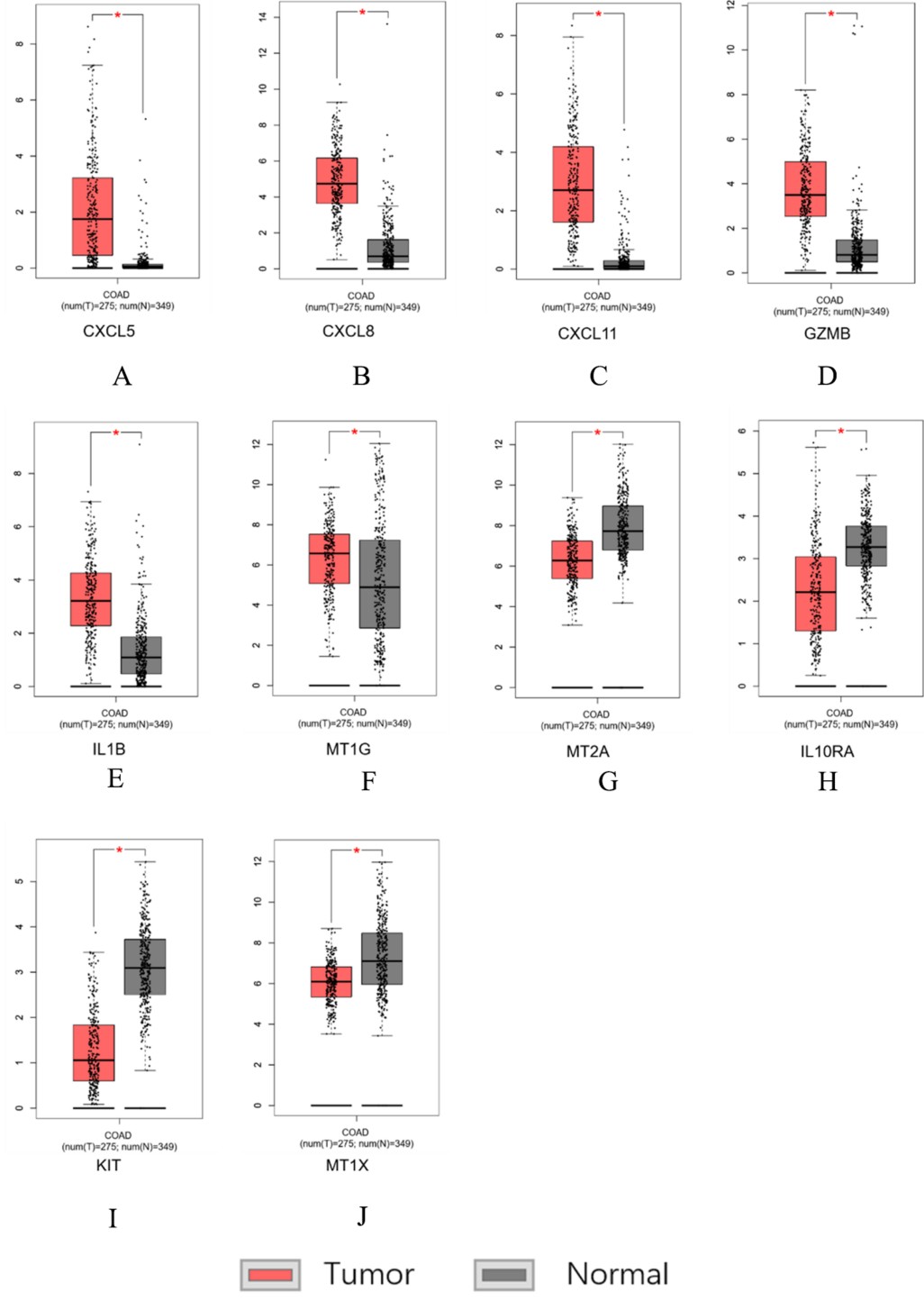

Figure 7 **Hub genes screened by TCGA-COAD data validation.** The expression level of hub genes in colorectal cancer tissues. The expression level of hub genes in colorectal cancer tissues was higher than those in normal tissues (*p*-value < 0.05). (A) CXCL5. (B) CXCL8. (C) CXCL11. (D) GZMB. (E) IL1B. (F) MT1G. The expression level of DEGs in colorectal cancer tissues was lower than normal tissues (indicates *p*-value < 0.05). (G) MT2A. (H) IL10RA. (I) KIT. (J) MT1X.

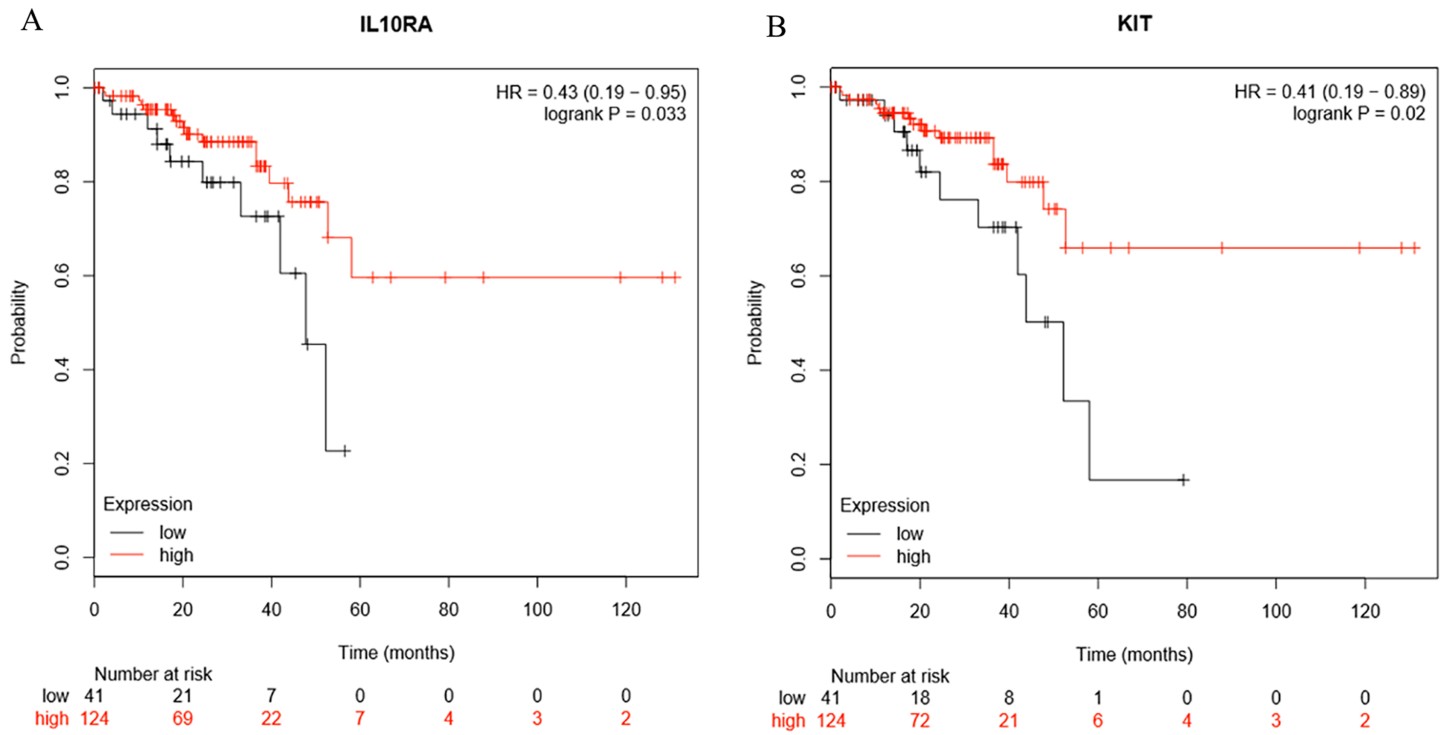

**Figure 8  The expression level of two hub genes was significantly associated with prognosis (*p*-value < 0.05).** The expression of the two hub genes selected increased and the overall survival time was significantly prolonged. (A) IL10RA. (B) KIT.

tendency to separate, with only a few being significantly concentrated. Expression changes occurring in one gene can interact with the linked genes and can affect the downstream biological functions. Finally, through survival analysis, we screened 10 genetic models including *MT1X, MT1G, MT2A, CXCL8, IL1B, CXCL5, CXCL11, IL10RA, GZMB, KIT*. Abnormal expression of these genes may influence the survival time of patients. The expression of *MT1G, CXCL8, IL1B, CXCL5, CXCL11, GZMB* in CRC tissues is higher than normal tissues. In contrast the expression of *MT1X, MT2A, IL10RA, KIT* was lower in cancer tissues as compared to normal tissues.

The researchers have found that these 10 hub genes are involved in cytokine-cytokine receptor interactions and in the receptor-ligand activity. Previous findings suggest that *MT1X* induces cell cycle arrest and apoptosis by inactivating NF-κB signaling in HCC (*Liu et al., 2018*). *MT1G* promotes new tumor suppressor activity in CRC tumor differentiation and indicates that MT and zinc signaling as new participants in colorectal differentiation (*Arriaga et al., 2017*). *MT1G* promotes methylation and tumor aggressiveness in prostate cancers and could serve as a marker for locally advanced disease (*Henrique et al., 2005*). The genetic polymorphisms in *MT2A* (rs10636 and rs28366003) increases the risk of breast cancer in Chinese Han population (*Liu et al., 2017*). In another study, overexpression of *CXCL8* induced cell proliferation, migration, and invasion of colon cancer LoVo cells, and *CXCL8* induced EMT via the PI3K/AKT/ NF-κB signaling axis was reported (*Shen et al., 2017*). Many articles have shown that IL1B

SNPs may be involved in the pathogenesis of *NSCLC* and thyroid cancer in Chinese population and can also be used as a new prognostic genetic biomarker for non-small cell lung cancer (*Li et al., 2019*, *2015*; *Perez-Ramirez et al., 2017*). Tumor-derived *CXCL5* can promote human CRC metastasis by activating ERK/Elk-1/Snail and AKT/GSK3β/β-catenin pathway, accelerate osteosarcoma growth and can serve as a biomarker for non-small cell carcinoma, etc. (*Roca et al., 2018*; *Wu et al., 2017*; *Zhao et al., 2017*). *CXCL11* is overexpressed in CRC tissues and cell lines, inhibiting *CXCL11* to significantly affect the CRC cell migration, invasion and EMT in vitro. In addition, down-regulation of *CXCL11* also reduces CRC growth and metastasis in vivo (*Gao et al., 2018*). *IL10* is a key anti-inflammatory cytokine inhibiting the pro-inflammatory responses of innate and adaptive immune cells. Spontaneous intestinal inflammation in *IL10* and IL10R-deficient mice is reported in the reference. Also patients with deleterious mutations in *IL10, IL10RA* or *IL10RB* develop severe enterocolitis in the first few months of life (*Shouval et al., 2014*). Moreover, the expression of *IL10* in CRC tissues was significantly higher than the healthy intestinal endothelial cells. The correlation between the expression of *IL10RA* and the proliferation index or clinical stage of CRC confirms the importance of *IL10RA* in the pathogenesis of CRC (*Zadka et al., 2018*). In NK cell-based anticancer therapy, activation of autophagy in hypoxic cells operates through selective degradation of the pro-apoptotic NK-derived serine protease GZMB/granzyme B and by blocking the NK-mediated target cell apoptosis. The autophagy targeting cancer cells promotes tumor regression by promoting the elimination of NK cells (*Viry et al., 2014*). Subpopulations of naive and memory B cells also express GZMB in mammary gland draining lymph nodes (*Arabpour et al., 2019*). Oncogenic signaling of Kit tyrosine kinase selectively occurs in the Golgi apparatus in the gastrointestinal stromal tumors and causing inhibition of the carcinogenesis by blocking the secretory transport of M-COPA in the gastrointestinal stromal tumors (*Obata et al., 2018*; *Obata et al., 2017*). Multiple reports have identified genes directly or indirectly involved in different biological pathways associated with CRC. The 10 genes identified in this study were further validated using a separate data set to verify if the 10 gene model can significantly affect the prognosis of CRC.

In practice, this 10-gene model can be used to roughly predict the prognosis. In order to further elucidate the relationship between DEGs and clinical outcomes in patients with CRC, additional mathematical analysis and modeling independent of standard clinical and pathological criteria needs to be conducted. Also, further prognostic gene models and cluster analysis of TNM (tumor, lymph node and metastasis) are needed to assess the independent nature of this model. At the same time, the identified genetic model requires further validation using qPCR and other laboratory methods.

## CONCLUSIONS

In summary, we screened DEGs based on GEO's gene expression profiling. GO enrichment analysis and KEGG pathway analysis were performed on these genes. The top 25% of the genes were screened for constructing a weighted co-expression network. Finally, hub genes were screened, and survival curves were generated. A separate data set

was used to verify the validity of the 10 gene model. The hub genes identified in this study can help to predict postoperative treatment and prognosis of CRC patients.

## ACKNOWLEDGEMENTS

The authors thank the reviewers for their helpful comments on our report.

### Funding
The authors received no funding for this work.

### Competing Interests
The authors declare that they have no competing interests.

### Author Contributions
- Jie Meng conceived and designed the experiments, performed the experiments, analyzed the data, prepared figures and/or tables, and approved the final draft.
- Rui Su performed the experiments, analyzed the data, prepared figures and/or tables, and approved the final draft.
- Yun Liao analyzed the data, prepared figures and/or tables, authored or reviewed drafts of the paper, contributed to cell lines and some consumables, and approved the final draft.
- Yanyan Li analyzed the data, authored or reviewed drafts of the paper, and approved the final draft.
- Ling Li conceived and designed the experiments, authored or reviewed drafts of the paper, and approved the final draft.

### Data Availability
The raw measurements are available in the Supplemental Files.

### Supplemental Information
Supplemental information for this article can be found online at http://dx.doi.org/10.7717/peerj.9633#supplemental-information.

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
