# Peer review of "Identification of 10 Hub genes related to the progression of colorectal cancer by co-expression analysis"

_PeerJ, doi:10.7717/peerj.9633_

## Round 0.1 · original submission · Minor Revisions

Please address issues pointed by the reviewers and revise the manuscript accordingly.

Reviewer 1 ·

Basic reporting

Meng et al.performed WGCNA to identify hub genes in the tumourigenesis of CRC. By combining data from TCGA and GEO, they identified ten hub genes (MT1X, MT1G, MT2A, CXCL8, IL1B, CXCL5, CXCL11, IL10RA, GZMB, KIT) and closely related to the progression of CRC. They performed a lot of computational analysis to validate their findings.

Experimental design

The experimental design is reasonable

Validity of the findings

1. In the result "GO functional and KEGG pathway analyses of mRNAs", could the author show some specific biological processes or pathways that are involved in the progression of CRC?
2. Please clarify your reasoning for selecting GSE28000 and GSE42284 for analysis from the GEO database.
3. This article identifies 10 related genes (MT1X, MT1G, MT2A, CXCL8, IL1B, CXCL5, CXCL11, IL10RA, GZMB, KIT) closely related to the progression of CRC were identified. Why IL10RA and KIT related to overall survival time?

Additional comments

1. The author's full text is too simple, and it is suggested to elaborate in detail, for example: introduction part (Please describe your rationale for selecting your candidate genes in the Introduction), result expression part.
2. I would suggest the authors to increase the figure resolution. It's hard to see clearly the texts in most figures.
3. In the result "GO functional and KEGG pathway analyses of mRNAs", could the author show some specific biological processes or pathways that are involved in the progression of CRC?
4. Please clarify your reasoning for selecting GSE28000 and GSE42284 for analysis from the GEO database.
5. There are some format errors in this article, which need to be modified.
6. This article identifies 10 related genes (MT1X, MT1G, MT2A, CXCL8, IL1B, CXCL5, CXCL11, IL10RA, GZMB, KIT) closely related to the progression of CRC were identified. Why IL10RA and KIT related to overall survival time?

Reviewer 2 ·

Basic reporting

The authors constructed a prognostic model based on the candidate ten genes by using the data set GSE28000 and GSE42284. WGCNA,co-expression network and the PPI network were used to screen hub genes. Survival analysis suggested that IL10RA and KIT were significantly associated with the prognosis of patients.Overall, the authors performed a lot of validation work for CRC.I still have some concerns.

Experimental design

The experimental design is reasonable

Validity of the findings

1.Why did the authors select GSE28000 and GSE42284 for further analysis?
2.The pre-processing procedures of the TCGA and GEO RNA-Seq datasets should be explained in detail.
3.The title of this article is“Identification of 10 Hub genes related to the
progression and prognosis of colorectal cancer by co-expression analysis”.Why survival analysis suggested that only IL10RA and KIT were significantly associated with the prognosis of patients, how about survival analysis of other eight genes?
4.I would suggest the authors to increase the figure resolution. It's hard to see clearly the texts in most figures.

Additional comments

1. Authors have done a sloppy job in writing the entire manuscript. English check/edits are required to make the article easy to understand.
2.The authors should re-check of language mistakes in the revised version.

Reviewer 3 ·

Basic reporting

I think the article is well written, the logic is clear, the research content is reliable, and the science is good, which basically meets the publication requirements of the magazine.

Experimental design

It's a very good study. And the aim of this study is clear. Methods described sufficient detail & information to replicate. Some researchers may interest in it.

Validity of the findings

This manuscript is well written and novely, I think that it has the potential for acceptance of peer j.

---

## Round 0.2 · accepted · Accept

Since all the critiques were adequately addressed and the manuscript was revised accordingly, the amended version is acceptable now.